# Asynchrony-Robust Collaborative Perception via Bird's Eye View Flow

**Sizhe Wei**[1][†]   **Yuxi Wei**[1][†]   **Yue Hu**[1]   **Yifan Lu**[1]
**Yiqi Zhong**[2]   **Siheng Chen**[1,3*]   **Ya Zhang**[1,3*]

[1] Cooperative Medianet Innovation Center, Shanghai Jiao Tong University
[2] University of Southern California    [3] Shanghai AI Laboratory
[1] `{sizhewei, wyx3590236732, 18671129361, yifan_lu}@sjtu.edu.cn`
`{sihengc, ya_zhang}@sjtu.edu.cn`
[2] `yiqizhon@usc.edu`

## Abstract

Collaborative perception can substantially boost each agent's perception ability by facilitating communication among multiple agents. However, temporal asynchrony among agents is inevitable in the real world due to communication delays, interruptions, and clock misalignments. This issue causes information mismatch during multi-agent fusion, seriously shaking the foundation of collaboration. To address this issue, we propose CoBEVFlow, an asynchrony-robust collaborative perception system based on bird's eye view (BEV) flow. The key intuition of CoBEVFlow is to compensate motions to align asynchronous collaboration messages sent by multiple agents. To model the motion in a scene, we propose BEV flow, which is a collection of the motion vector corresponding to each spatial location. Based on BEV flow, asynchronous perceptual features can be reassigned to appropriate positions, mitigating the impact of asynchrony. CoBEVFlow has two advantages: (i) CoBEVFlow can handle asynchronous collaboration messages sent at irregular, continuous time stamps without discretization; and (ii) with BEV flow, CoBEVFlow only transports the original perceptual features, instead of generating new perceptual features, avoiding additional noises. To validate CoBEVFlow's efficacy, we create IRregular V2V(IRV2V), the first synthetic collaborative perception dataset with various temporal asynchronies that simulate different real-world scenarios. Extensive experiments conducted on both IRV2V and the real-world dataset DAIR-V2X show that CoBEVFlow consistently outperforms other baselines and is robust in extremely asynchronous settings. The code is available at https://github.com/MediaBrain-SJTU/CoBEVFlow.

## 1 Introduction

Multi-agent collaborative perception allows agents to exchange complementary perceptual information through communication. This can overcome inherent limitations of single-agent perception, such as occlusion and long-range issues. Recent studies have shown that collaborative perception can substantially boost the performance of perception systems [1–5] and has great potential for a wide range of real-world applications, such as multi-robot automation system [6, 7], vehicle-to-everything-communication-aided autonomous driving[8, 9] and multi-UAVs (unmanned aerial vehicles) [10–12]. As an emerging area, the study of collaborative perception has many challenges to be tackled, such as high-quality datasets [13–15], model-agnostic and task-agnostic formulation [16] and the robustness to pose errors [17] and adversarial attacks [18, 19].

---

[†]Equal contribution.    [*]Corresponding author.

37th Conference on Neural Information Processing Systems (NeurIPS 2023).

However, a vast majority of existing works do not seriously account for the harsh realities of real-world communication among agents, such as congestion, heavy computation, interruptions, and the lack of calibration. These factors introduce delays or misalignments that severely impact the reliability and quality of information exchange among agents. Some prior works have touched upon the issue of communication latency. For instance, V2VNet [4] and V2X-ViT [5] incorporated the delay time as an input for feature compensation. However, they only account for a single frame without leveraging historical frames, making them inadequate for high-speed scenarios (above 20m/s) or high-latency scenarios (above 0.3s) scenarios. Meanwhile, SyncNet [20] uses historical features to predict the complete feature map at the current timestamp [21]. Nevertheless, this RNN-based method assumes equal time intervals for its input, causing failures when delays are irregular. Overall, previous works have not addressed the issues raised by common irregular time delays, rendering existing collaborative perception systems can never reach their full potential in real-world scenarios.

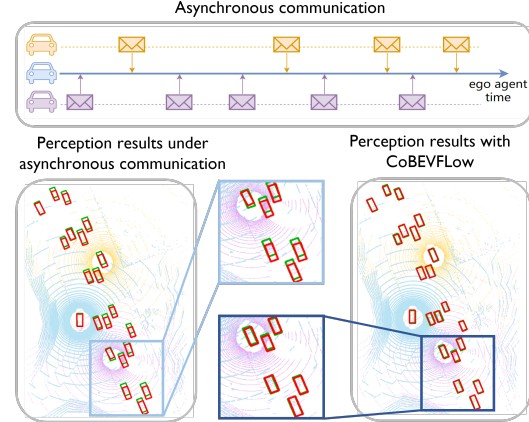

Figure 1: Illustration of asynchronous collaborative perception and the perception result w.o./w. CoBEVFlow. Red boxes are detection results and green boxes are the ground-truth.

To fill the research gap, we specifically formulate a setting of *asynchronous* collaborative perception; see Fig. 1 for a visual demonstration. Here asynchrony indicates that the time stamps of the collaboration messages from other agents are not aligned and the time interval of two consecutive messages from the same agent is irregular. Due to the universality and inevitability of temporal asynchrony in practical applications, handling this setting is critical to the further development of collaborative perception. To address this, we propose CoBEVFlow, an asynchrony-robust collaborative perception system based on bird's eye view (BEV) flow. The key idea is to align perceptual information from other agents by compensating relative motions. Specifically, CoBEVFlow uses historical frames to estimate a BEV flow map, which encodes the motion information in each grid cell. Using the BEV flow map, CoBEVFlow can reassign asynchronous perceptual features to appropriate spatial locations, which aligns perceptual features on the time dimension, mitigating the impact caused by asynchrony. The proposed CoBEVFlow has two major advantages: i) CoBEVFlow can handle asynchronous collaboration messages sent at irregular, continuous timestamps without discretization; ii) CoBEVFlow better adheres to the essence of compensation, which is to move features to the designated spatiotemporal positions. Not like SyncNet[20] that regenerates features, this motion-guided position adjustment fundamentally prevents introducing extra noises to the features.

When validating the effectiveness of CoBEVFlow, we noticed that there is no appropriate collaborative perception dataset that contains asynchronous samples. To facilitate research on asynchronous collaborative perception, we create IRregular V2V(IRV2V), the first synthetic asynchronous collaborative perception dataset with irregular time delays, simulating various real-world scenarios. Sec. 5 show the experiment results and analysis on both IRV2V and a real-world dataset DARI-V2X[14]. Results show that CoBEVFlow consistently achieves the best compensation performance across various latencies. When the expected latency on the IRV2V dataset is set to 500ms, CoBEVFlow outperforms other methods by more than 18.9%. In the case of a 300ms latency with an additional 200ms disturbance, the decrease in AP@0.50 is only 0.25%.

## 2 Related work

### 2.1 Collaborative Perception

Factors such as limited sensor fields of view and physical environmental occlusions can negatively impact perception tasks for individual agents[22, 23]. To address the aforementioned challenges, collaborative perception based on multi-agent systems has emerged as a promising solution [2, 3, 5, 1, 24–26]. It can enhance the performance of perception tasks by leveraging the exchange of information among different agents within the same scenario. V2VNet uses multi-round message passing via graph neural networks to achieve better perception and prediction performance[4]; DiscoNet adopts knowledge distillation to take advantage of both early and intermediate collaboration[9]; V2X-ViT proposes a heterogeneous multi-agent attention module to aggregate information from heterogeneous

agents; Where2comm[1] introduces a spatial confidence map which achieves pragmatic compression and improves perception performance with limited communication bandwidth.

As unideal communication is an inevitable issue that negatively impacts the performance and application of collaborative perception, some methods have been studied for robust collaborative perception. V2VNet[4] utilizes a convolutional neural network to learn how to compensate for communication delay by taking time information and relative pose as input; V2X-ViT[5] designs a Delay-aware Positional Encoding module to learn the influence caused by latency, but these methods do not consider the historical temporal information for compensation. SyncNet[20] uses historical multi-frame information and compensates for the current time by Conv-LSTM[21], but its compensation for the whole feature map leads noises to the feature channels, and RNN-based framework can not handle temporal irregular inputs. This work formulates asynchronous collaborative perception, which considers real-world communication asynchrony.

## 2.2 Time-Series Forecasting

Time series analysis aims to extract temporal information of variables, giving rise to numerous downstream tasks. Time series forecasting leverages historical sequences to predict observations of variables in future time periods. Classical methods for time series forecasting include the Fourier transform[27], autoregressive models[28], and Kalman filters[29]. In the era of deep learning, a plethora of RNN-based and attention-based methods have emerged to model sequence input and address time series prediction problems[30–32]. The issue of sampling irregularity, which may arise due to physical interference or device issues, calls for irregularity-robust time series analysis systems. Approaches such as mTAND[33], IP-Net[34], and DGM$^2$[35] employ imputation-based techniques to tackle irregular sampling, which estimate the observation or hidden embedding of series at regular timestamps and then apply the methods designed for regular time series analysis, while SeFT[36] and Raindrop[37] utilize operations that are insensitive to sampling intervals for processing irregularly sampled data. In this work, we consider each collaboration message as one irregular sample and use irregular time series forecasting to estimate the BEV flow map for asynchronous feature alignment.

## 3 Problem Formulation

Consider $N$ agents in a scene, where each agent can send and receive collaboration messages from other agents, and store $k$ historical frames of messages. For the $n$th agent, let $\mathcal{X}_n^{t_n^i}$ and $\mathcal{Y}_n^{t_n^i}$ be the raw observation and the perception ground-truth at time current $t_n^i$, respectively, where $t_n^i$ is the $i$-th timestamp of agent $n$, and $\mathcal{P}_{m\to n}^{t_m^j}$ be the collaboration message sent from agent $m$ at time $t_m^j$. The key of the asynchronous setting is that the timestamp of each collaboration message $t_n^i \in \mathbb{R}$ is a continuous value, those messages from other agents are not aligned, $t_m^i \neq t_n^i$, and the time interval between two consecutive timestamps $t_n^{i-1} - t_n^i$ is irregular. Therefore, each agent has to encounter collaboration messages from other agents sent at arbitrary times. Then, the task of asynchronous collaborative perception is formulated as:

$$\max_{\theta, \mathcal{P}} \quad \sum_{n=1}^{N} g\left(\widehat{\mathbf{Y}}_n^{t_n^i}, \mathbf{Y}_n^{t_n^i}\right) \tag{1}$$

$$\text{subject to} \quad \widehat{\mathbf{Y}}_n^{t_n^i} = \mathbf{c}_\theta(\mathcal{X}_n^{t_n^i}, \{\mathcal{P}_{m\to n}^{t_m^j}, \mathcal{P}_{m\to n}^{t_m^{j-1}}, \cdots, \mathcal{P}_{m\to n}^{t_m^{j-k+1}}\}_{m=1}^N),$$

where $g(\cdot, \cdot)$ is the perception evaluation metric, $\widehat{\mathbf{Y}}_n^{t_n^i}$ is the perception result of agent $n$ at time $t_n^i$, $\mathbf{c}_\theta(\cdot)$ is the collaborative perception network with trainable parameters $\theta$, and $t_m^{j-k+1} < t_m^{j-k+2} < \cdots < t_m^j \leq t_n^i$. Note that: i) when the collaboration messages from other agents are all aligned and the time interval between two consecutive timestamps is regular; that is, $t_m^i = t_n^i$ for all agent's pairs $m, n$, and $t_n^i - t_n^{i-1}$ is a constant for all agents $n$, the task degenerates to standard well-synchronized collaborative perception; and ii) when the collaboration messages from other the agents are not aligned, yet the time interval between two consecutive timestamps is regular; that is, $t_m^i \neq t_n^i$ and $t_n^i - t_n^{i-1}$ is a constant, the task degenerates to the setting in SyncNet [20].

Given such irregular asynchrony, the performances of collaborative perception systems would be significantly degraded since features from asynchronized timestamps differ from the actual current features, and using asynchronized features may contain erroneous information during the perception process. In the next section, we will introduce CoBEVFlow to address this critical issue.

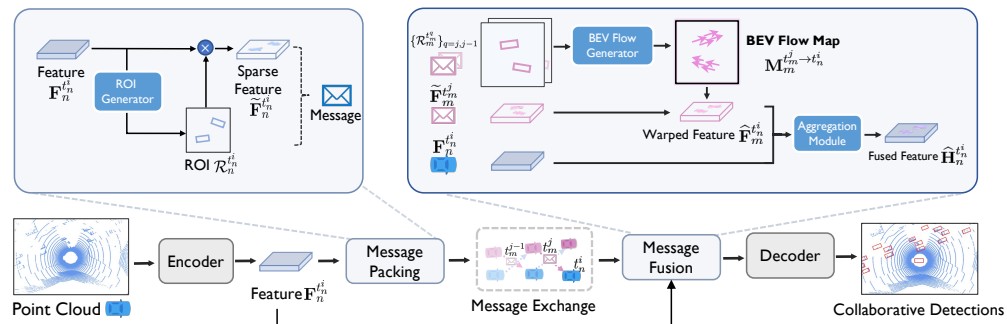

Figure 2: System overview. Message packing process prepares ROI and sparse features as the message for efficient communication and BEV flow map generation. Message fusion process generates and applies BEV flow map for compensation, and fuses the features at the current timestamp from all agents.

# 4 CoBEVFlow: Asynchrony-Robust Collaborative Perception System

This section proposes an asynchrony-robust collaborative perception system, CoBEVFlow. Figure 2 overviews the overall scheme of CoBEVFlow. We introduce the overall framework of the CoBEVFlow system in Sec. 4.1. The details of three key modules of CoBEVFlow can be found in Sec. 4.2-4.4. Sec. 4.5 demonstrates the training details and loss functions of the whole system.

## 4.1 Overall architecture

The problem of asynchrony results in the misplacements of moving objects in the collaboration messages. That is, the collaboration messages from multiple agents would record various positions for the same moving object. The proposed CoBEVFlow addresses this issue with two key ideas: i) we use a BEV flow map to capture the motion in a scene, enabling motion-guided reassigning asynchronous perceptual features to appropriate positions; and ii) we generate the region of interest(ROI) to make sure that the reassignment only happens to the areas that potentially contain objects. By following these two ideas, we eliminate direct modification of the features and keep the background feature unaltered, effectively avoiding unnecessary noise in the learned features.

Mathematically, let the $n$-th agent be the ego agent and $\mathbf{X}_n^{t_n^i}$ be its raw observation at the $i$-th timestamp of agent $n$, denoted as $t_n^i$. The proposed asynchrony-robust collaborative perception system CoBEVFlow is formulated as follows:

$$\mathbf{F}_n^{t_n^i} = f_{\text{enc}}(\mathbf{X}_n^{t_n^i}), \tag{2a}$$

$$\widetilde{\mathbf{F}}_n^{t_n^i}, \mathcal{R}_n^{t_n^i} = f_{\text{roi\_gen}}(\mathbf{F}_n^{t_n^i}), \tag{2b}$$

$$\mathbf{M}_m^{t_m^j \to t_n^i} = f_{\text{flow\_gen}}(t_n^i, \{\mathcal{R}_m^{t_m^q}\}_{q=j-k+1,j-k+2,\cdots,j}), \tag{2c}$$

$$\widehat{\mathbf{F}}_m^{t_n^i} = f_{\text{warp}}(\widetilde{\mathbf{F}}_m^{t_m^j}, \mathbf{M}_m^{t_m^j \to t_n^i}), \tag{2d}$$

$$\widehat{\mathbf{H}}_n^{t_n^i} = f_{\text{agg}}(\widetilde{\mathbf{F}}_n^{t_n^i}, \{\widehat{\mathbf{F}}_m^{t_n^i}\}_{m \in \mathcal{N}_n}), \tag{2e}$$

$$\widehat{\mathbf{Y}}_n^{t_n^i} = f_{\text{dec}}(\widehat{\mathbf{H}}_n^{t_n^i}), \tag{2f}$$

where $\mathbf{F}_n^{t_n^i} \in \mathbb{R}^{H \times W \times D}$ is the BEV perceptual feature map of agent $n$ at timestamp $t_n^i$ with $H, W$ the size of BEV map and $D$ the number of channels; $\mathcal{R}_n^{t_n^i}$ is the set of region of interest (ROI); $\widetilde{\mathbf{F}}_n^{t_n^i} \in \mathbb{R}^{H \times W \times D}$ is the sparse version of $\mathbf{F}_n^{t_n^i}$, which only contains features inside $\mathcal{R}_n^{t_n^i}$ and zero-padding outside; $\mathbf{M}_m^{t_m^j \to t_n^i} \in \mathbb{R}^{H \times W \times 2}$ is the $m$-th agent's the BEV flow map that reflects each grid cell's movement from timestamp $t_m^j$ to timestamp $t_n^i$, $\{\mathcal{R}_m^{t_m^q}\}_{q=j-k+1,j-k+2,\cdots,j}$ indicates historical $k$ ROI sets sent by agent $m$; $\widehat{\mathbf{F}}_m^{t_n^i} \in \mathbb{R}^{H \times W \times D}$ is the realigned feature map from the $m$-th agent's at timestamp $t_n^i$ after motion compensation; $\widehat{\mathbf{H}}_n^{t_n^i} \in \mathbb{R}^{H \times W \times D}$ is the aggregated features from all of the agents; $\mathcal{N}_n$ is the collaboration neighbors of the $n$-th agent; and $\widehat{\mathbf{Y}}_n^{t_n^i}$ is the final output of the system.

Step 2a extracts BEV perceptual feature from observation data. Step 2b generates the ROIs for each feature map, enabling BEV flow generation in Step 2c. Now all the agents exchange their messages,

including $\widetilde{\mathbf{F}}_n^{t_n^i}$ and the $\mathcal{R}_n^{t_n^i}$. Step 2c generate the BEV flow map $\mathbf{M}_m^{t_m^j \to t_n^i}$ by leveraging historical ROIs from the same agent. Step 2d gets the estimated feature map by applying the BEV flow map to reassign the asynchronized features. Step 2e aggregates the feature maps of all agents. Finally, Step 2f outputs the final perceptual results.

Note that i) Steps 2a-2b are done before communication. Steps 2c-2f are performed after receiving the message from others. During the communication process, both sparse perceptual features and the ROI set are sent to other agents, which is communication bandwidth friendly; and ii) CoBEVFlow adopts the feature representations in bird's eye view (BEV), where the feature maps of all agents are projected to the same global coordinate system, avoiding complex coordinate transformations and supporting easier cross-agent collaboration.

The proposed asynchronous collaborative perception system has three advantages: i) CoBEVFlow can deal with asynchronous collaboration messages sent at irregular, continuous timestamps without discretization; (ii) with BEV flow, CoBEVFlow only transports the original perceptual features, instead of generating new perceptual features, causing additional noises; and (iii) CoBEVFlow promotes robustness to asynchrony by introducing minor communication cost (ROI set).

We now elaborate on the details of Steps 2b-2e in the following subsections.

## 4.2 ROI generation

Given the perceptual feature map of an agent, Step 2b aims to generate a set of spatial regions of interest (ROIs) for the areas that possibly contain objects. Each ROI indicates one potential object's region in the scene. The intuition is that foreground objects are the only ones that move, while the background remains static. Therefore, using ROI can enable the subsequent BEV flow map to concentrate on critical regions and simplify the computation of the BEV flow map.

To implement, we use the structure of the object detection decoder to produce the ROIs. Given agent $m$'s perceptual feature map at timestamp $t_m^j$, $\mathbf{F}_m^{t_m^j}$, the corresponding detection result is obtained as:

$$\mathbf{O}_m^{t_m^j} = \Phi_{\text{roi\_gen}}(\mathbf{F}_m^{t_m^j}) \in \mathbb{R}^{H \times W \times 7}, \tag{3}$$

where $\Phi_{\text{roi\_gen}}(\cdot)$ is the ROI generation network with detection decoder structure, and each element $(\mathbf{O}_m^{t_m^j})_{h,w} = (c, x, y, h, w, \cos\alpha, \sin\alpha)$ represents one detected ROI with its class confidence, position, size, and orientation. We threshold the class confidence, apply non-max suppression, and obtain a set of detected boxes, whose occupied spaces form the set of ROIs, $\mathcal{R}_m^{t_m^j}$.

Based on this ROI set, we can also get a binary mask $\mathbf{H} \in \mathbb{R}^{H \times W}$, whose values inside ROIs are 1 and the others are 0. We then get the sparse feature map $\widetilde{\mathbf{F}}_m^{t_m^j} = \mathbf{F}_m^{t_m^j} \odot \mathbf{H}$, which only contains the features within ROIs. Then, agent $m$ packs its sparse feature map $\widetilde{\mathbf{F}}_m^{t_m^j}$ and the ROI set $\mathcal{R}_m^{t_m^j}$ as the message and sends out for collaboration.

## 4.3 BEV flow map generation

After receiving collaboration messages from other agents at various timestamps, Step 2c aims to generate the BEV flow map to correct feature misalignment due to asynchrony. The proposed BEV flow map encodes the motion vector of each spatial location. The main idea of obtaining this BEV flow is to associate correlated ROIs based on a sequence of messages sent by the same collaborator. In this step, each ROI is regarded as an instance that has its own attributes generated by Step 2b. After the ROI association, we are able to compute the motion vector and further estimate the positions where the corresponding object would appear at a certain timestamp. The generation of the BEV flow map includes two key steps: adjacent timestamp' ROI matching and BEV flow estimation.

**Adjacent frames' ROI matching.** The purpose of adjacent frames' ROI matching is to match the ROIs in two consecutive messages sent by the same agent. The matched ROIs are essentially the same instance perceived at different timestamps. This module contains three processes: cost matrix construction, greedy matching, and post-processing. We first construct a cost matrix $\mathbf{C} \in \mathbb{R}^{o_1 \times o_2}$, where $o_1$ and $o_2$ are the numbers of ROIs in the two frames to be matched. Each element $\mathbf{C}_{p,q}$ is the matching cost between ROI $p$ in the earlier frame and ROI $q$ in the later frame. To determine the value of $\mathbf{C}_{p,q}$, we define the vicinity of the front and rear directions as a feasible angle range for matching. We set $\mathbf{C}_{p,q} = d_{p,q}$ when $q$ is within the feasible angle range of $p$, otherwise $\mathbf{C}_{p,q} = +\infty$, where $d_{p,q}$ is the Euclidean distance between the center of ROI $p$ and $q$. We then use the greedy matching

strategy to search the paired ROIs. For each row $p$, we search the $q$ with minimum $\mathbf{C}_{p,q}$, and match $p$, $q$ as a pair. To avoid invalid matching, we further post-process matched pairs by removing those with excessively large values of $C_{p,q}$. Through these processes, we can get the matched ROI pairs in adjacent frames. For a sequence of frames, we can track each ROI's multiple locations across frames.

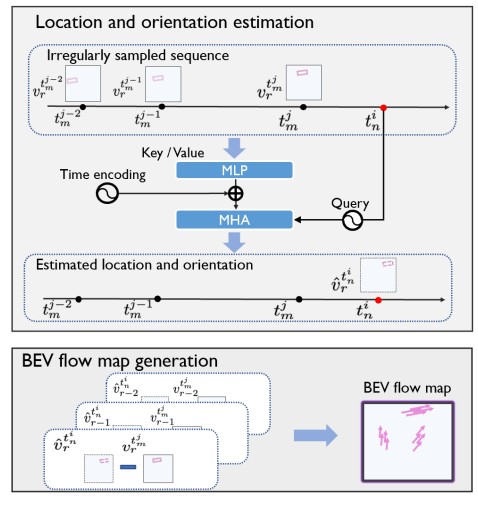

Figure 3: The process of the BEV flow estimation.

**BEV flow estimation.** We now retrieve each ROI's historical locations at a series of irregular timestamps. In this module, we use those irregular tracklets to predict the location and orientation of ROIs at the ego agent's current timestamp $t_n^i$ and generate the corresponding BEV flow map $\mathbf{M}_m^{t_m^j \to t_n^i}$. To formulate the $r$-th ROI's irregular tracklet perceived by $m$-th agent, we extract the motion-related attributes (i.e. location and orientation) from the $r$-th ROI's attributes set at each timestamp. Let $\mathbf{V}_{m,r} = \{\mathbf{v}_r^{t_m^j}, \mathbf{v}_r^{t_m^{j-1}}, \cdots, \mathbf{v}_r^{t_m^{j-k+1}}\}$ be a historical sequence of the $r$-th ROI's attributes sent by the $m$-th agent, where $\mathbf{v}_r^{t_m^j} = (x_r^{t_m^j}, y_r^{t_m^j}, \alpha_r^{t_m^j})$, with $(x_r^{t_m^j}, y_r^{t_m^j})$ is the 2D BEV center position and $\alpha_r^{t_m^j}$ is the orientation. Note that $\mathbf{V}_{m,r}$ is an irregularly sampled sequence due to time asynchrony. Fig 3 shows this process.

Based on $\mathbf{V}_{m,r}$, we now predict $\mathbf{v}_r^{t_n^i}$, which is the location and orientation of the $r$-th ROI at the ego agent's current timestamp. Unlike the common motion estimation, we need to handle an irregularly sampled sequence. To enable the irregularity-compatible motion estimation method, the information of the timestamp should be taken into account. Here we propose to use traditional trigonometric functions [38] for timestamp encoding, by which we map the continuous-valued timestamp $t$ into its corresponding time code $\mathbf{u}(t)$ through:

$$(\mathbf{u}(t))_{2e} = \sin(\frac{t}{10000^{2e/d}}), \quad (\mathbf{u}(t))_{2e+1} = \cos(\frac{t}{10000^{2e/d}}), \tag{4}$$

where $e$ is the index of temporal encoding. The timestamp information now can be input into the estimation process along with the irregularly sampled sequence and make the estimation process capable of irregularity-compatible motion estimation. We implement the estimation process using multi-head attention(MHA). The query of MHA is the time code of the target timestamp $t_n^i$, and the key and value both are the sum of the features of the irregularly sampled sequence and its corresponding time code set $\mathbf{U}_k$:

$$\widehat{\mathbf{v}}_r^{t_n^i} = \mathrm{MHA}(\mathbf{u}(t_n^i), \mathrm{MLP}(\mathbf{V}_{m,r}) + \mathbf{U}_k, \mathrm{MLP}(\mathbf{V}_{m,r}) + \mathbf{U}_k), \tag{5}$$

where $\widehat{v}_r^{t_n^i}$ is the estimation of ROI's location and orientation $v_r^{t_n^i}$, $\mathrm{MLP}(\cdot)$ is the encoding function of the irregular historical series $\mathbf{V}_m^r$, and $\mathrm{MHA}(\cdot)$ is the multi-head attention for temporal estimation.

With the estimated location and orientation of ROIs in the ego agent's current timestamp along with the ROIs' sizes predicted by Step 2b, we calculate the motion vector at each grid cell by an affine transformation of the associated ROI's motion, constituting the whole BEV flow map $\mathbf{M}_m^{t_m^j \to t_n^i} \in \mathbb{R}^{H \times W \times 2}$. Note that the grid cells outside ROIs are zero-padded.

Compared to Syncnet [20] that uses RNNs to handle regular communication latency, the generated BEV flow map has two benefits: i) it handles irregular asynchrony via the attention-based estimation with appropriate time encoding; and ii) it facilitates the motion-guided feature warping, which avoids regenerating the entire feature channels.

### 4.4 Feature warp and aggregation

The BEV flow map $\mathbf{M}_m^{t_m^j \to t_n^i}$ is applied on the sparse feature map $\widetilde{\mathbf{F}}_m^{t_m^j}$, which implements Step 2d. The features at each grid cell are moved to the estimated position based on $\mathbf{M}_m^{t_m^j \to t_n^i}$. The warping

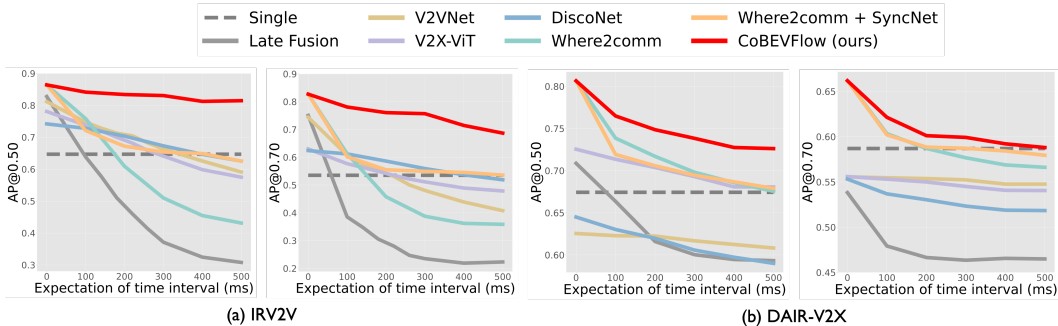

(a) IRV2V                                  (b) DAIR-V2X

Figure 4: Comparison of the performance of CoBEVFlow and other baseline methods under the expectation of time interval from 0 to 500ms. CoBEVFlow outperforms all the baseline methods and shows great robustness under any level of asynchrony on both two datasets.

process is: $\widetilde{\mathbf{F}}_m^{t_n^i}\left[h + \mathbf{M}_m^{t_m^j \to t_n^i}[h, w, 0], w + \mathbf{M}_m^{t_m^j \to t_n^i}[h, w, 1]\right] = \mathbf{F}_m^{t_m^j}[h, w]$. After adjusting each non-ego agent's feature map, these estimated feature maps and ego feature map are aggregated together by an aggregation function $f_{\mathrm{agg}}(\cdot)$, which implements Step 2e. The fusing function can be any common fusion operation. All our experiments adopt Multi-scale Max-fusion.

## 4.5 Training details and loss function

To train the overall system, we supervise three tasks: ROI generation, flow estimation, and the final fusion detector. As mentioned before, the functionality of the ROI generator and final fusion detector share the same architecture but do not share the parameters. During the training process, the ROI generator and flow estimation module are trained separately, and later the final fusion detector is trained with the pre-trained two modules. Common loss functions in detection tasks: cross entropy loss and weighted smooth L1 loss are used for the classification and regression of ROI generation and final fusion detector, and MSE loss is used for flow estimation.

## 5 Experimental Results

We propose the first asynchronous collaborative perception dataset and conduct extensive experiments on both simulated and real-world scenarios. The task of the experiments on the two datasets is point-cloud-based object detection. The detection performance was evaluated using Average Precision (AP) at Intersection-over-Union (IoU) thresholds of 0.50 and 0.70.

### 5.1 Datasets

**IRregular V2V(IRV2V).** To facilitate research on asynchrony for collaborative perception, we simulate the first collaborative perception dataset with different temporal asynchronies based on CARLA [39], named IRregular V2V(IRV2V). We set 100ms as ideal sampling time interval and simulate various asynchronies in real-world scenarios from two main aspects: i) considering that agents are unsynchronized with the unified global clock, we uniformly sample a time shift $\delta_s \sim \mathcal{U}(-50, 50)$ms for each agent in the same scene, and ii) considering the trigger noise of the sensors, we uniformly sample a time turbulence $\delta_d \sim \mathcal{U}(-10, 10)$ms for each sampling timestamp. The final asynchronous time interval between adjacent timestamps is the summation of the time shift and time turbulence. In experiments, we also sample the frame intervals to achieve large-scale and diverse asynchrony. Each scene includes multiple collaborative agents ranging from 2 to 5. Each agent is equipped with 4 cameras with a resolution of $600 \times 800$ and a 32-channel LiDAR. The detection range is 281.6m $\times$ 80m. It results in 34K images and 8.5K LiDAR sweeps. See more details in the Appendix.

**DAIR-V2X.** DAIR-V2X [14] is a real-world collaborative perception dataset. There is one ego agent and one roadside unit in each frame. All frames are captured from real scenarios at 10 Hz with 3D annotations. Lu et al. [17] complemented the missing annotations outside the camera view on the vehicle side to cover a 360-degree view detection. We adopt the complemented annotations[17] and set the perceptual range to $x \in [-100.8\mathrm{m}, +100.8\mathrm{m}]$, $y \in [-40\mathrm{m}, +40\mathrm{m}]$.

### 5.2 Quantitative evaluation

**Benchmark comparison.** The baseline methods include late fusion, DiscoNet[9], V2VNet[4], V2X-ViT[5] and Where2comm[1]. The red dashed line represents single-agent detection without

Table 1: Detection performance on IRV2V and DAIR-V2X[14] dataset with pose noises following Gaussian distribution in the testing phase.

| Dataset | IRV2V | | | | | DAIR-V2X | | | | |
|---|---|---|---|---|---|---|---|---|---|---|
| Noise Level $\sigma_t/\sigma_r(m/\circ)$ | 0.0/0.0 | 0.1/0.1 | 0.2/0.2 | 0.3/0.3 | 0.4/0.4 | 0.0/0.0 | 0.1/0.1 | 0.2/0.2 | 0.3/0.3 | 0.4/0.4 |
| Model / Metric | AP@0.50 ↑ | | | | | | | | | |
| V2X-ViT | 0.641 | 0.626 | 0.627 | 0.625 | 0.619 | 0.693 | 0.692 | 0.545 | 0.685 | 0.681 |
| Where2comm | 0.510 | 0.411 | 0.411 | 0.411 | 0.411 | 0.702 | 0.693 | 0.679 | 0.658 | 0.643 |
| Where2comm+SyncNet | 0.654 | 0.653 | 0.652 | 0.651 | 0.648 | 0.711 | 0.692 | 0.583 | 0.579 | 0.671 |
| CoBEVFlow (ours) | **0.831** | **0.820** | **0.815** | **0.802** | **0.781** | **0.738** | **0.743** | **0.732** | **0.723** | **0.703** |
| Model / Metric | AP@0.70 ↑ | | | | | | | | | |
| V2X-ViT | 0.511 | 0.504 | 0.502 | 0.504 | 0.501 | 0.545 | 0.545 | 0.545 | 0.685 | 0.543 |
| Where2comm | 0.388 | 0.323 | 0.312 | 0.302 | 0.293 | 0.577 | 0.577 | 0.561 | 0.658 | 0.543 |
| Where2comm+SyncNet | 0.549 | 0.550 | 0.545 | 0.538 | 0.527 | 0.587 | 0.583 | 0.579 | 0.570 | **0.567** |
| CoBEVFlow (ours) | **0.757** | **0.730** | **0.686** | **0.628** | **0.570** | **0.599** | **0.593** | **0.579** | **0.571** | 0.560 |

collaboration. We also consider the integration of SyncNet[20] with Where2comm[1], which presents the SOTA method Where2comm[1] with resistance to time delay. All methods use the same feature encoder based on PointPillars[40]. To simulate temporal asynchrony, we sample the frame intervals of received messages with binomial distribution to get random irregular time intervals. Fig. 4 shows the detection performances (AP@IoU=0.50/0.70) of the proposed CoBEVFlow and the baseline methods under varying levels of temporal asynchrony on both IRV2V and DAIR-V2X, where the $x$-axis is the expectation of the time interval of delay of the latest received information and interval between adjacent frames and $y$-axis the detection performance. Note that, when the $x$-axis is at 0, it represents standard collaborative perception without any asynchrony. We see that i) the proposed CoBEVFlow achieves the best performance in both simulation and real-world datasets at all asynchronous settings. On the IRV2V dataset, CoBEVFlow outperforms the best methods by 23.3% and 35.3% in terms of AP@0.50 and AP@0.70, respectively, under a 300ms interval expectation. Similarly, under a 500ms interval expectation, we achieve 30.3% and 28.2% improvements, respectively. On the DAIR-V2X dataset, CoBEVFlow still performs best. ii) CoBEVFlow demonstrates remarkable robustness to asynchrony. As shown by the red line in the graph, CoBEVFlow exhibits a decrease of only 4.94% and 14.0% in AP@0.50 and AP@0.70, respectively, on the IRV2V dataset under different asynchrony. These results far exceed the performance of single-object detection, even under extreme asynchrony.

**Trade-off between detection results and communication cost.** CoBEVFlow allows agents to share only sparse perceptual features and the ROI set, which is communication bandwidth-friendly. Figure 5 compares the proposed CoBEVFlow with the previous methods in terms of the trade-off between detection performance (AP@0.50/0.70) and communication bandwidth under asynchrony. We adopt the same asynchrony settings mentioned before and choose 300ms as the expectation of the time interval. We see: i) CoBEVFlow consistently outperforms the state-of-the-art communication efficient solution, where2comm, as well as the other baselines in the setting of asynchrony; ii) as the communication volume increases, the performance of CoBEVFlow continues to improve

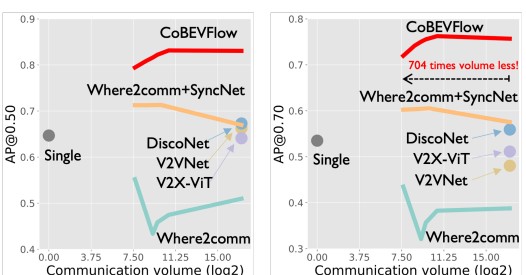

Figure 5: Trade-off between detection performance (AP@0.50/0.70) and communication bandwidth under asynchrony (expected 300ms latency) on IRV2V dataset. CoBEVFlow outperforms even with a much smaller communication volume.

steadily, while the performance of where2comm and where2comm+SyncNet fluctuates due to improper information transformation caused by asynchrony.

**Robustness to pose error.** We conduct experiments to validate the performance under the impact of both asynchrony and pose error. To simulate the pose error, we add Gaussian noise $\mathcal{N}(0, \sigma_t)$ on $x, y$ and $\mathcal{N}(0, \sigma_r)$ on $\theta$ during the inference phase, where $x, y, \theta$ are 2D centers and yaw angle of accurate global poses. Our pose noise setting follows the Gaussian distribution with a mean of 0m, a standard deviation of 0m-0.5m, a mean of $0°$ and a standard deviation of $0° - 0.5°$. This experiment is conducted under the expectation of time interval is 300ms to simulate the time asynchrony. We compare our CoBEVFlow and other baseline methods including V2X-ViT[5], Where2comm[1]

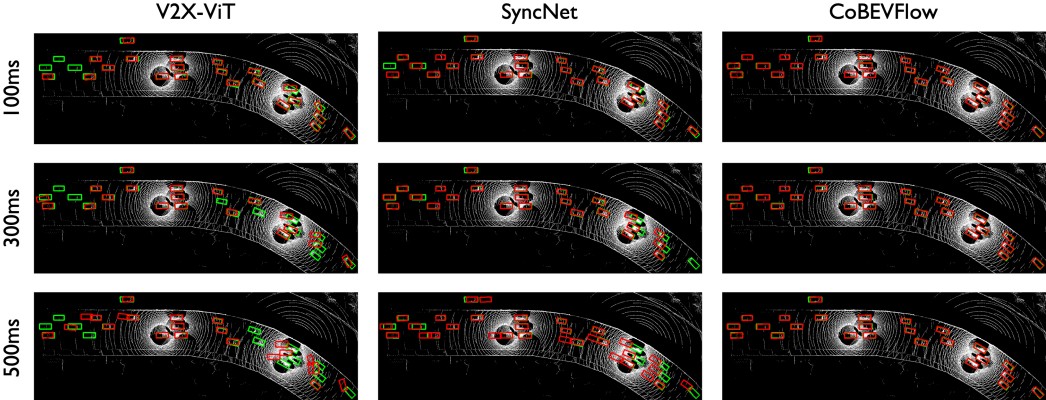

Figure 6: Visualization of detection results for V2X-ViT, SyncNet, and CoBEVFlow with the expectation of time intervals are 100, 300, and 500ms on IRV2V dataset. CoBEVFlow qualitatively outperforms the others under different asynchrony. Red and green boxes denote detection results and ground-truth respectively.

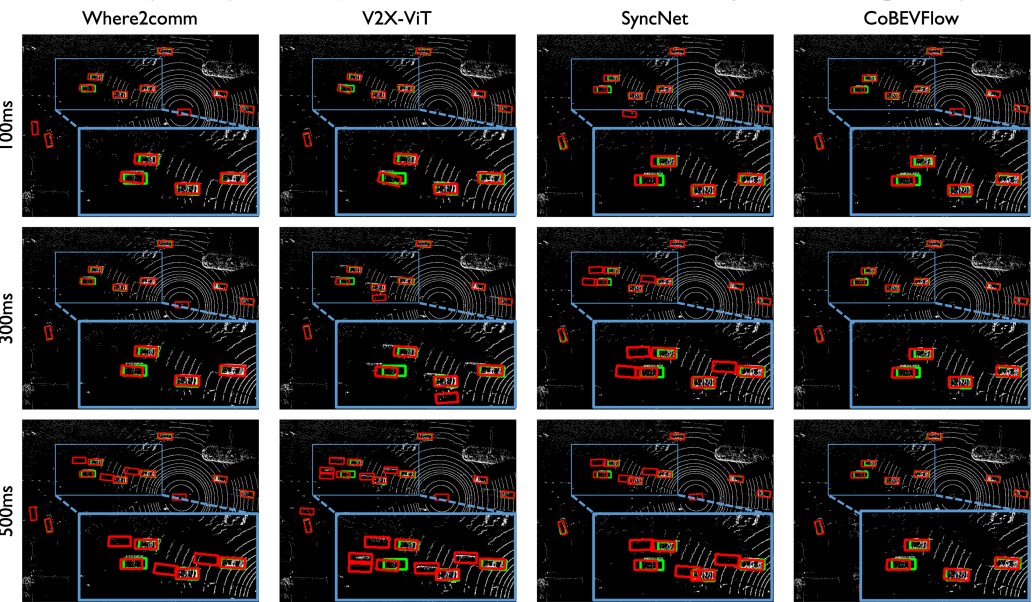

Figure 7: Visualization of detection results for Where2comm, V2X-ViT, SyncNet, and our CoBEVFlow with the expectation of time intervals are 100, 300, and 500ms on the DAIR-V2X dataset. Red and green boxes denote detection results and ground-truth respectively.

and SyncNet[20]. Table 1 shows the results on IRV2V and DAIR-V2X[14] dataset. We see that **CoBEVFlow still performs well even when both pose errors and time asynchrony appear**. CoBEVFlow consistently outperforms other methods across all noise settings on the IRV2V dataset. In the case of noise levels of 0.4/0.4, our approach achieves 0.133 and 0.043 improvement over SyncNet.

### 5.3 Qualitative evaluation

**Visualization of detection results.** We illustrate the detection results of V2X-ViT, SyncNet, and CoBEVFlow at three asynchrony levels on the IRV2V dataset in figure 6 and the DAIR-V2X dataset in figure 7. The expectations of time intervals are 100, 300, and 500ms. The red box represents the detection result and the green box represents the ground truth. V2X-ViT shows significant deviations in collaborative perception under asynchrony, while SyncNet shows poor compensation due to introducing noise in feature regeneration and irregularity-incompatible design. The third row shows the results of CoBEVFlow, which achieves precise compensation and outstanding detections.

**Visualization of BEV flow map.** Figure 8 visualizes the feature map before/after compensation of CoBEVFlow in Plot(a)(b), the corresponding flow map in Plot(c), and matching, detection results

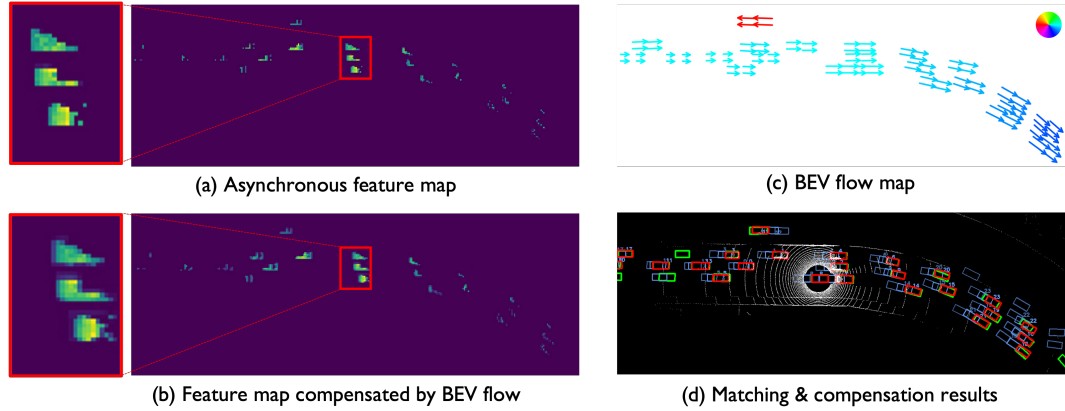

(a) Asynchronous feature map

(c) BEV flow map

(b) Feature map compensated by BEV flow

(d) Matching & compensation results

Figure 8: Visualization of compensation with CoBEVFlow on IRV2V dataset. In subfigure(d), green boxes are the objects' ground truth locations, blue boxes are the detection results based on the historical asynchronous features and red boxes are the detection results after compensation. CoBEVFlow achieves precise matching and compensation with the BEV flow map and mitigates the negative impact of asynchrony to a great extent.

after compensation in Plot(d). The green boxes in Plot (d) are the ground truth, the blue boxes are the historical detections with the matched asynchronous ROIs and the red boxes are the compensated detections. We see that the BEV flow map can be precisely estimated and is beneficial for perceptual feature alignment. The compensated detection results are more accurate than the uncompensated ones.

## 5.4 Ablation Study

We conduct ablation studies on the IRV2V dataset. Table 2 assesses the effectiveness of the proposed operations, including time encoding, the object(feature/detected box) to warp, and our ROI matcher. We see that: i) time encoding encodes the irregular continuous timestamp and makes the estimation more accurate; ii) Warping the features outperforms warping the boxes directly a lot, which means the operation for features shows superiority over the operation for the detection results; and iii) our ROI matcher shows more proper matching results than traditional Hungarian[41] matching.

Table 2: Ablation Study on IRV2V dataset. Time encoding(TE), BEV flow on features, and the proposed matcher all improve the performance.

| Modules | | | AP@0.50 / AP@0.70 ↑ | |
|---|---|---|---|---|
| TE | Warped by Flow | Matcher | 300ms | 500ms |
| | Box | Hungarian | 0.724 / 0.473 | 0.644 / 0.388 |
| ✓ | Box | Hungarian | 0.747 / 0.595 | 0.668 / 0.438 |
| ✓ | Box | Ours | 0.764 / 0.611 | 0.571 / 0.399 |
| ✓ | Feature | Hungarian | 0.779 / 0.690 | 0.739 / 0.614 |
| ✓ | Feature | Ours | **0.831 / 0.757** | **0.815 / 0.687** |

## 6 Conclusion and limitation

We formulate the asynchrony collaborative perception task, which considers various unideal factors that may cause communication latency or information misalignments during collaborative communication. We further propose CoBEVFlow, a novel asynchrony-robust collaborative perception framework. The core idea of CoBEVFlow is BEV flow, which is a collection of the motion vector corresponding to each spatial location. Based on BEV flow, asynchronous perceptual features can be reassigned to appropriate positions, mitigating the impact of asynchrony. Comprehensive experiments show that CoBEVFlow achieves outstanding performance under all settings and far superior robustness with asynchrony.

**Limitation and future work.** The current work focuses on addressing the asynchrony problem in collaborative perception. It is evident that effective prediction can compensate for the negative impact of temporal asynchrony in collaborative perception. Moreover, the generated flow can also be utilized not only for compensation but also for prediction. In the future, we expect more works on exploring the ROI-based flow generation design for collaborative perception and prediction tasks.

**Acknowledgment.** This research is supported by NSFC under Grant 62171276 and the Science and Technology Commission of Shanghai Municipal under Grant 21511100900, 22511106101, and 22DZ2229005.

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

# A   IRregular V2V (IRV2V)

To facilitate the research on asynchrony for collaborative perception, we use CARLA [39] (under MIT license) to simulate IRregular V2V (IRV2V) dataset, which is the first collaborative perception dataset with multiple asynchronies.

**Asynchronous data collection.** The number of collaborative vehicles in a scene ranges from 2 to 5. Each collaborative vehicle is equipped with 4 cameras for $360°$ view, a 32-channel LiDAR, and GPS/IMU sensors. The ideal sample interval of the sensor is 100ms. Due to different asynchronous factors, collaborative messages have asynchronous timestamps. There is a time offset $\delta_s \sim \mathcal{U}(-50, 50)$ms at the sampling starting point of non-ego vehicles. And all non-ego vehicles' collaborative messages are sampled with time turbulence $\delta_d \sim \mathcal{U}(-10, 10)$ms. Sensing information at each timestamp of each agent contains 4 camera images with resolution $600 \times 800$, and 32-channel LiDAR points.

**Data size.** Assuming the model requires the use of information from the past 10 frames, our dataset consists of a total of 8,449 collaborative samples, which include 8,449 point cloud inputs and 33,796 RGB images. We have split the dataset into training, validation, and testing sets, which contain 5,445, 994, and 2,010 samples, respectively.

**Data analysis.** Figure 9 presents some statistical analysis results regarding the IRV2V dataset. The IRV2V dataset contains a total of 1,564,033 vehicles, with an average of 48.302 vehicles per scene. It should be noted that the figure only displays the distribution of vehicles with speeds greater than 1 km/h. Considering real-world scenarios, there are around 1,203,793 moving vehicles in the dataset. Plot (a) illustrates the distribution of moving vehicles with different speeds across all samples, ranging from 1 to 105 km/h, with an average speed of 25.586 km/h, which achieves around 15km/h faster compared to the majority of vehicles in the V2X-Sim[15]. Plot (b) shows the distribution of the total number of vehicles per sample in the dataset, with the maximum number of vehicles being 113.

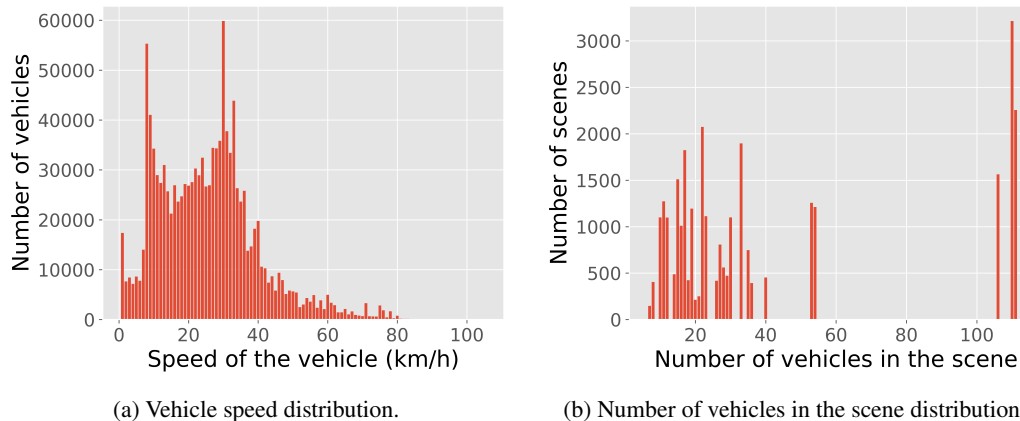

(a) Vehicle speed distribution.    (b) Number of vehicles in the scene distribution.

Figure 9: Data distribution of IRV2V dataset. (a) shows the speed distribution of moving vehicles; (b) shows the vehicle numbers distribution.

# B   Detailed information about experimental settings

**Implement details.** We conduct experiments on LiDAR-based part of IRV2V and DAIR-V2X[14] dataset. Our feature encoder is PointPillars[40] based. And our backbone follows the setting in CoAlign[17]. The difference is that we change the fusion method from self-attention to max-fusion. We conduct training for a total of 60 epochs, starting with an initial learning rate of 2e-3. Subsequently, at the 10th and 20th epochs, the learning rate decreases to 10% of its previous value. For IRV2V dataset, we set the lidar range as $x \in [-140.8, +140.8]$m, $y \in [-40, +40]$m. The voxel size is $h = w = 0.4$m. The feature map's size is $H = 200, W = 704$. For DAIR-V2X dataset, we set the lidar range as $x \in [-100.8, +100.8]$m, $y \in [-40, +40]$m. The voxel size is $h = w = 0.4$m. The feature map's size is $H = 200, W = 504$.

**Communication volume.** Our communication volume is the same as Where2comm[1]. For CoBEVFlow, we control the communication volume by adjusting the maximum number of generated ROIs. Specifically, the average number of voxels contained in each ROI region is 40, and we limit the maximum number of generated ROIs to $K_{\|\mathcal{R}\|}$. Correspondingly, we modify the information exchange in Where2comm to include the top $40 \times K_{\|\mathcal{R}\|}$ blocks based on their scores on the spatial confidence map. In practical scenarios, the actual communication volume is influenced by factors such as the feature dimension and floating-point precision. To simplify the expression, we uniformly represent the communication volume using the logarithm to the base 2 of the voxel count. The communication volume is

$$\log_2 \left( 40 \times K_{\|\mathcal{R}\|} \right), \tag{6}$$

where $K_{\|\mathcal{R}\|}$ is the maximum number of generated ROIs, and 40 is the average number of voxels in each ROI.

## C  Benchmarks

We conduct extensive experiments on current collaborative perception methodologies. Table 3 and Table 4 present the detection performance under the expectation of time interval from 0 to 500ms on IRV2V and DAIR-V2X[14] respectively, which correspond to the numerical results shown in Figure 4 in the main text. We see that CoBEVFlow consistently achieves significant improvements over previous methods on both datasets and the leading gap is bigger when the expectation of the time interval is higher.

Table 3: Performance of CoBEVFlow and other baseline methods under the expectation of time interval from 0 to 500ms on IRV2V dataset. CoBEVFlow outperforms all the baseline methods and shows great robustness under any level of asynchrony.

| Expectation of interval (ms) | 0 | 100 | 200 | 300 | 400 | 500 |
|---|---|---|---|---|---|---|
| Model / Metric | AP@0.50 ↑ | | | | | |
| Single | 0.647 | | | | | |
| Late Fusion | 0.828 | 0.638 | 0.478 | 0.371 | 0.324 | 0.308 |
| V2VNet | 0.811 | 0.747 | 0.710 | 0.663 | 0.626 | 0.591 |
| V2X-ViT | 0.781 | 0.737 | 0.692 | 0.641 | 0.598 | 0.575 |
| DiscoNet | 0.742 | 0.728 | 0.704 | 0.673 | 0.647 | 0.625 |
| Where2comm | 0.864 | 0.758 | 0.609 | 0.510 | 0.455 | 0.431 |
| Where2comm+SyncNet | 0.864 | 0.721 | 0.672 | 0.654 | 0.649 | 0.625 |
| CoBEVFlow (ours) | **0.864** | **0.841** | **0.834** | **0.831** | **0.812** | **0.815** |
| Model / Metric | AP@0.70 ↑ | | | | | |
| Single | 0.535 | | | | | |
| Late Fusion | 0.751 | 0.385 | 0.285 | 0.235 | 0.219 | 0.223 |
| V2VNet | 0.744 | 0.607 | 0.540 | 0.480 | 0.439 | 0.408 |
| V2X-ViT | 0.630 | 0.577 | 0.545 | 0.511 | 0.489 | 0.479 |
| DiscoNet | 0.624 | 0.612 | 0.586 | 0.559 | 0.537 | 0.519 |
| Where2comm | 0.827 | 0.613 | 0.458 | 0.388 | 0.362 | 0.359 |
| Where2comm+SyncNet | 0.827 | 0.602 | 0.555 | 0.549 | 0.545 | 0.536 |
| CoBEVFlow (ours) | **0.827** | **0.781** | **0.761** | **0.757** | **0.714** | **0.687** |

We extended our experiments to include the V2XSet[5] dataset. V2XSet is a simulated dataset where each scenario involves at most one roadside unit and 2 to 4 vehicles as collaborative objects. The outcomes of these experiments are summarized in Table 5. Under time delays of 300ms and 500ms, the AP@0.70 scores achieved by CoBEVFlow are 0.776 and 0.713 respectively. These values surpass the best baseline methods by 17.0% and 10.6% respectively, and are notably higher than the results of single-object detection(0.556). This once again underscores CoBEVFlow's ability to maintain high levels of collaborative perception performance in scenarios involving temporal asynchrony.

Table 4: Performance of CoBEVFlow and other baseline methods under the expectation of time interval from 0 to 500ms on DAIR-V2X[14] dataset. CoBEVFlow outperforms all the baseline methods and shows great robustness under any level of asynchrony.

| Expectation of interval (ms) | 0 | 100 | 200 | 300 | 400 | 500 |
|---|---|---|---|---|---|---|
| Model / Metric | AP@0.50 ↑ | | | | | |
| Single | 0.674 | | | | | |
| Late Fusion | 0.709 | 0.664 | 0.616 | 0.600 | 0.595 | 0.593 |
| V2VNet | 0.626 | 0.623 | 0.622 | 0.617 | 0.612 | 0.608 |
| V2X-ViT | 0.725 | 0.714 | 0.704 | 0.693 | 0.681 | 0.681 |
| DiscoNet | 0.645 | 0.630 | 0.620 | 0.606 | 0.597 | 0.590 |
| Where2comm | 0.807 | 0.739 | 0.603 | 0.698 | 0.686 | 0.676 |
| Where2comm+SyncNet | 0.807 | 0.719 | 0.706 | 0.695 | 0.687 | 0.679 |
| CoBEVFlow (ours) | **0.807** | **0.765** | **0.749** | **0.738** | **0.728** | **0.726** |
| Model / Metric | AP@0.70 ↑ | | | | | |
| Single | 0.587 | | | | | |
| Late Fusion | 0.538 | 0.479 | 0.467 | 0.464 | 0.466 | 0.465 |
| V2VNet | 0.556 | 0.555 | 0.554 | 0.552 | 0.548 | 0.548 |
| V2X-ViT | 0.556 | 0.553 | 0.550 | 0.545 | 0.541 | 0.541 |
| DiscoNet | 0.553 | 0.537 | 0.530 | 0.523 | 0.519 | 0.518 |
| Where2comm | 0.662 | 0.603 | 0.587 | 0.577 | 0.569 | 0.566 |
| Where2comm+SyncNet | 0.662 | 0.602 | 0.588 | 0.587 | 0.584 | 0.580 |
| CoBEVFlow (ours) | **0.662** | **0.621** | **0.601** | **0.599** | **0.592** | **0.588** |

Table 5: We conduct experiments on the V2XSet dataset, comparing the performance of CoBEVFlow and other baseline methods under different expected time intervals. CoBEVFlow outperforms all the baseline methods and shows great robustness under any level of asynchrony.

| Expectation of interval (ms) | 0 | 300 | 500 |
|---|---|---|---|
| Model / Metric | AP@0.50/AP@0.70 ↑ | | |
| Single | 0.720 / 0.556 | | |
| Late Fusion | 0.881/0.751 | 0.460/0.307 | 0.442/0.333 |
| V2VNet | 0.906/0.781 | 0.581/0.334 | 0.524/0.341 |
| V2X-ViT | 0.912/0.751 | 0.712/0.529 | 0.642/0.499 |
| Where2comm | 0.901/0.847 | 0.677/0.542 | 0.612/0.511 |
| Where2comm+SyncNet | 0.901/0.847 | 0.801/0.663 | 0.781/0.645 |
| CoBEVFlow (ours) | 0.901/0.847 | **0.871/0.776** | **0.841/0.713** |

