# OpenReview forum: "Asynchrony-Robust Collaborative Perception via Bird's Eye View Flow"
_NeurIPS.cc/2023/Conference — NeurIPS 2023 poster_

### Official Review · Reviewer_vzpF · 2023-06-27

**Soundness:** 3 good
**Presentation:** 3 good
**Contribution:** 3 good
**Rating:** 7
**Confidence:** 5

**Summary:**

This article proposes a robust detection algorithm to address the issue of detection errors caused by asynchronous information transmission in multi-agent collaborative perception tasks. For late fusion, a robust prediction algorithm is designed using a BEV flow map generation algorithm, which provides a method to align perception information from different time frames. Additionally, the authors claim to have introduced the first asynchronous collaborative perception dataset and tested the algorithm on this dataset as well as the DAIR-V2X dataset.

**Strengths:**

This paper addresses an important research problem in multi-agent collaborative perception tasks and proposes a structurally concise and effective 3D detection framework. By utilizing BEV flow, the framework significantly reduces the bandwidth consumption in information transmission among different agents. To overcome the challenge of low matching accuracy in information fusion during asynchronous collaborative perception, the paper introduces a BEV flow map that can align neighboring frames of vehicles. By predicting the BEV flow, the framework aligns detection results with time intervals. Additionally, the paper presents a benchmark dataset specifically designed for asynchronous collaborative perception tasks, which contributes to the advancement of research in this field

**Weaknesses:**

The writing in this paper can be further improved to enhance readers' understanding of the formulas and methods described. In the main experiments, such as the benchmark comparison in the quantitative evaluation, it would be helpful to include a table that presents the comparative results for easier reference.

**Questions:**

1. The experiments in this paper are not comprehensive enough, as they only validate the proposed methods on the dataset introduced in this paper, but lack testing on datasets such as V2XSet[1].
2. The research motivation in this paper focuses on the time intervals between timestamps, but it seems insufficient to be a major concern. Even if the time intervals for information transmission between agents are inconsistent, the prediction algorithm can still rely on predicting algorithms combined with several consecutive data transmissions. In other words, does the algorithm's performance significantly differ when the time intervals are the same or different?
3. In the section "Adjacent frames' ROI matching," the author uses the Euclidean distance between the same ROI in two consecutive data transmissions from the same agent as the association criterion. However, when there is high vehicle density in the environment, this can significantly impact the association algorithm. How is this issue addressed?
4. Also in the section "Adjacent frames' ROI matching," how is the "feasible angle range" defined in line 213?

[1]. Xu, et al. V2X-ViT: Vehicle-to-Everything Cooperative Perception with Vision Transformer. ECCV 2022.


**Limitations:**

The proposed algorithm in this paper seems to be affected by complex vehicle scenarios.

---

> ### Author Rebuttal · Authors · 2023-08-09
>
> ## W1: Enhancing Clarity of Formulas, Methods, and Experiments
>
> Thank you for your feedback. We will revise the corresponding expressions in the final version for better understanding. We have provided data tables in the appendix corresponding to the main text, which can be referred to more accurately.
>
> ---
>
> ## Q1: Dataset Validation and Comparison
>
> In the main text, the proposed method is validated on two datasets, our new dataset **IRV2V** and **DAIR-V2X**. Note that DAIR-V2X is the first real-world collaborative perception dataset, which is collected from real road scenes. In this dataset, there is one vehicle and one roadside unit. The communication process takes place between the vehicle and the roadside unit. The dataset is collected from real-world scenarios, encompassing a wide range of conditions including clear, rainy, and foggy days, as well as daytime, nighttime, urban roads, and highways. It is extensively employed in collaborative perception research. The experiments conducted on DAIR-V2X are mainly shown in Figure 4(b), Table 4(Appendix), and Figure 8(Appendix).  Figure.4(b) in the main text and Table 4 in the appendix show that CoBEVFlow outperforms the best methods by 5.73% and 2.04% in terms of AP@0.50 and AP@0.70, respectively, under a 300ms interval expectation. The red line represents our method, CoBEVFlow, and it can be observed that our method consistently outperforms others. In Appendix Figure 8, we present the visualization of detection results on the DAIR-V2X dataset with an expected interval of 300ms. The red boxes indicate the predicted results, while the green boxes represent the ground truth. In the presence of temporal asynchrony, our method's predictions closely match the ground truth, resulting in fewer false positives. These results demonstrate that CoBEVFlow continues to achieve superior performance in real-world scenarios.
>
> The V2XSet dataset mentioned by the reviewer is a simulation dataset that does not explicitly consider latency issues. We show the **additional experimental results** on V2XSet in **Table 1** of the global response pdf. CoBEVFlow outperforms the best SOTA methods by 8.74%, and 17.04% for AP@0.50 and AP@0.70 respectively under the expectation of time intervals is 300ms. The findings obtained from experiments on the V2XSet dataset align with the conclusions drawn from experiments on the other two datasets in this paper. CoBEVFlow demonstrates its ability to alleviate the impact of temporal asynchrony and maintain robustness in the presence of temporal asynchrony.
>
> ---
>
> ## Q2: Impact of Time Intervals
>
> Different time intervals lead to significantly different performance for methods that rely on temporal information and assume known fixed time intervals. Under the assumption of regular sampling, the temporal variation information between continuous frames is fixed, and the model does not need to consider or extract the specific time interval between consecutive timestamps. When facing inputs with random different time intervals, the model may mistakenly perceive them as having a fixed interval, leading to incorrect analysis of temporal change speed and incorrect compensation for specific delays. Syncnet, discussed in the paper, is such a method. To validate this point, we trained Syncnet (which is an LSTM-based method) with a fixed time interval of 100ms, while using different time intervals during the inference process. We compensate based on two frames of historical information, fixing the delay of the first frame at 500ms. And we vary the time interval between the second and the first frame at 500, 600, 700, 800, and 900ms. In the table, $\Delta t$ represents the difference between two time intervals. The experimental comparison is shown in Table 2 in the global response PDF. Syncnet cannot cope well with varying time intervals, its performance is significantly affected. However, our method remains highly robust in the face of time irregularities.
>
> ---
>
> ## Q3: Addressing Association Algorithm Impact in Dense Environments
>
> For such situations, we can leverage object tracking methods. During matching, not only the distance but also the similarity of the features covered within the ROI range are considered to solve this problem. However, in most cases, such situations are very rare in all our datasets. Our current matching method still has very accurate matching results even when vehicles are relatively dense, so there is not an urgent need for this kind of requirement.
>
> ---
>
> ## Q4: "Feasible Angle Range" in ROI Matching
>
> The "feasible angle range" is actually the range of angles to which a vehicle may move during regular driving. It is generally a certain angular range with the vehicle's front or rear as the centerline. In our experiments, we set it uniformly to plus or minus 45 degrees.

---

> > ### Comment · Reviewer_vzpF · 2023-08-14
> >
> > The author addressed my doubts, so I will raise the score of the paper from 6 to 7.

---

### Official Review · Reviewer_NCLy · 2023-07-05

**Soundness:** 3 good
**Presentation:** 4 excellent
**Contribution:** 4 excellent
**Rating:** 8
**Confidence:** 5

**Summary:**

The paper proposes CoBEVFlow, a new system for collaborative 3D perception that can handle temporal asynchrony among multiple agents. CoBEVFlow compensates for motion to align asynchronous collaboration messages and has two advantages: it can handle irregular time stamps without discretization and only transports original perceptual features. The authors validate CoBEVFlow's efficacy with a newly proposed synthetic dataset IRV2V and an existing real-world dataset (DAIR-V2X), showing that CoBEVFlow outperforms other methods and is robust in extremely asynchronous settings. Overall, the paper presents a novel approach to collaborative perception that improves collaboration among multiple agents.

**Strengths:**

- The paper contributes to the formulation of the asynchrony collaborative perception task and proposes a novel asynchrony-robust collaborative perception framework. Compared with the existing methods to solve the delay, the proposed benchmark is more challenging and closer to real-world scenarios.

- CoBEVFlow has two advantages over other methods. Firstly, it can handle asynchronous collaboration messages sent at irregular, continuous time stamps without discretization. Secondly, it only transports the original perceptual features, avoiding additional noise and generating new perceptual features.

- Comprehensive experiments conducted on both IRV2V and a real-world dataset, DAIR-V2X, show that CoBEVFlow consistently outperforms other baselines and is robust in extremely asynchronous settings.

- The writing in this paper is good, and the figures and charts are clear.

**Weaknesses:**

- The training process requires three stages, which are complex and time-consuming, which limits practical application scenarios.

- The collaboration performance is constrained by the performance of the ROI generator.

**Questions:**

- Time complexity analysis of the proposed ROI matching method and the Hungarian matching method.

- Based on the information provided, it is unclear why the performance of Where2comm+SyncNet increases with a decrease in communication volume in Figure 5. Further investigation is needed to understand the reasons behind this phenomenon.

**Limitations:**

Yes, they have addressed them.

---

> ### Author Rebuttal · Authors · 2023-08-09
>
> ## W1: Training Process Complexity and Practical Application
> The training process does require three stages, but it is not time-consuming and does not limit practical application scenarios for two reasons.
> 1. First, each of the three modules is lightweight. The training of the individual detection module, and the finetuning of the collaborative components, are common to all collaborative perception tasks. The stage of training the prediction module is both lightweight and fast. The entire prediction training stage takes only a matter of minutes and does not entail significant additional time consumption during the inference time.
> 2. Additionally, the inference process is also not time-consuming. The time required for inference remains unaffected by the intricacies of the training stages. It is noteworthy that the inference time doesn't exhibit a mere increase as a result of its nature of multi-stage training. To support our point, We examine the inference time increment compared to the Where2Comm approach, which does not account for temporal asynchrony, our method introduces only a marginal 3% increment in time consumption during the inference time with proper implementation optimization. Furthermore, we observed that when compared to SyncNet, another method that equally considers delay compensation, our approach remarkably demonstrates a reduction of 33% in time consumption.
> ---
> ## W2: Performance Dependency on ROI Generator
> That's a good question. We agree with the reviewer's insightful comment. But we want to emphasize two points:
> 1. ROI generator generally provides reliable ROIs. The performance of the ROI generator is strongly correlated with the performance of the single-object detection. There are many mature and powerful single-object detection methods available now. In most cases, we can easily find an excellent single-object detection method and adopt it as the ROI generator with minor modifications. According to our experiments, the ROI generator can easily achieve commendable results even using the most common 3D detectors, such as PointPillar, which we use to generate the experiments results table in the manuscript. We will add this information to the revised version.
> 2. When the ROI generator fails, our collaboration performance is still better than baselines. In extreme cases where the ROI generator performs poorly, it indicates that it is difficult to find a sufficiently good single-object detection method. In this case, all collaborative perception methods would struggle to perform well. However, even if only a portion of the ROIs are accurate and effective, our method can still compensate for them, thereby achieving stronger robustness to time asynchrony compared to other methods.
> ---
>
> ## Q1: Time Complexity Analysis of ROI Matching Methods
>
> The theoretical time complexity of our matching scheme is **$O(n)$**, while the general theoretical time complexity of the Hungarian algorithm is **$O(n^3)$**, so our scheme has a significant advantage in terms of theoretical time complexity. Given that about 50 ROIs are typically encountered during the detection process, our method only consumed 12ms to complete the matching task, while the Hungarian algorithm took about 16ms. In practice, the time consumption for matching is a very small part of the overall inference process and can basically be ignored.
>
> ---
> ## Q2: Investigating SyncNet's Performance Increase with Decreased Communication Volume
> The reason that the performance of Where2comm+SyncNet increases with a decrease in communication volume might be that SyncNet applies a Conv-LSTM-based compensation strategy which struggles to accurately conduct motion compensation when features have lower quality. With the increase in communication volume, features with lower confidence are more likely to be transmitted compared to situations when the communication volume is lower. It indicates that the overall qualities of the features become lower when the communication volume increase. Therefore, the motion compensation capability of SyncNet tends to be worse in such situations and results in worse performance. To support our point, Figure 1 in the global response PDF visualizes the detection results of CoBEVFlow and Syncnet under different bandwidths. We can observe that as the bandwidth increases, the number of false positives in Syncnet's detection results noticeably grows, showing some misalignment with the ground truth.

---

### Official Review · Reviewer_iS3b · 2023-07-06

**Soundness:** 3 good
**Presentation:** 3 good
**Contribution:** 3 good
**Rating:** 6
**Confidence:** 5

**Summary:**

This work points out that there is a time delay among agents and the delay period is not fixed. Thus, when these agents share their environment perception information in BEV, there will be an uneven spatial mismatch. To address the aforementioned problem, this work constructs a benchmark and proposes a strategy to align the BEV feature of various agents by predicting BEV flow map.

**Strengths:**

1.	This work finds a new problem and develops a corresponding benchmark. This studied problem is interesting and meaningful.
2.	The proposed strategy for tackling the asynchronous perception problem is easy to understand and makes sense.


**Weaknesses:**

1.	The proposed algorithm predicts the BEV flow estimation rather than movement speed, so the communication delay among agents should be known. How to know that in practical applications.
2.	In the developed benchmark, there is a random time turbulence. How to tackle this turbulence?


**Questions:**

not.

**Limitations:**

Please see the weakness section.

---

> ### Author Rebuttal · Authors · 2023-08-09
>
> ## W1: Communication Delay in Practical Applications
> In practical applications, it is straightforward to know the communication delay because different vehicles can easily acquire a unified world timestamp.
> 1. A unified world timestamp is easily obtainable for agents. Several existing technologies can achieve this capability, such as GPS systems, network time servers, and online time services.
> 2. When vehicles send information, they also transmit the corresponding world timestamp. Thus, the information received by the ego agent comes with its specific timestamp, which allows the computation of the exact time interval and delay. Given the globally synchronized time stamps, the time intervals or delays can be easily calculated.
> ---
> ## W2: Tackle Random Time Turbulence
> Here we assume that "Tackle this turbulence" mentioned by the review means the process of mitigating the effect of temporal asynchrony. Here are the responses:
> 1. Regarding temporal asynchrony in real-world scenarios: The 'turbulence' is a crucial factor of the temporal asynchrony in the data. Correspondingly, in real scenarios, this turbulence emulates a variety of potential unfavorable factors, including but not limited to situations where the sensors or associated hardware are unable to handle their tasks, resulting in data collection not proceeding entirely at the predetermined frequency; disruptions like illumination, electromagnetic waves, and unstable hardware and software drivers that could interfere with the precise moments when the sensor collects information. These adverse factors all contribute to turbulence at the time of data collection. The method we propose in our paper can address asynchrony caused by any factor.
> 2. About the datasets: IRV2V is a synthetic dataset created by adjusting the sampling times to simulate temporal asynchrony. The sampling times are continuous and more complex, which better reflects real-world applications. The other dataset we used,  DAIR-V2X, is a real dataset sampled at 10Hz. We simulate temporal asynchrony in situations like communication delays by selecting non-aligned perception information.
> 3. About the baseline methods: State-of-the-art collaborative perception methods like late fusion, DiscoNet, Where2comm, etc., do not consider the impact of temporal asynchrony. In our testing, we directly use asynchronous collaboration information as inputs to these models. SyncNet addresses temporal delays but assumes fixed time intervals between input information. In our experiments, we directly use irregularly spaced perception information as input, which proves to be challenging for these methods to handle effectively.
>
> In summary, CoBEVFlow addresses the challenges of temporal asynchrony in real-world scenarios, and our experiments (Figure 4) show that it outperforms other methods that do not explicitly consider such 'turbulence'. Please let me know if you have any further questions.

---

### Official Review · Reviewer_ZUTg · 2023-07-07

**Soundness:** 3 good
**Presentation:** 2 fair
**Contribution:** 3 good
**Rating:** 6
**Confidence:** 3

**Summary:**

The paper introduces CoBEVFlow, a new asynchrony-robust collaborative 3D perception system designed to enhance multi-agent perception abilities. The method compensates motions to align asynchronous collaboration messages from different agents, aiding in dealing with real-world issues such as communication delays. The authors developed BEVflow to reassign asynchronous perceptual features to appropriate positions and thus mitigating the impact of asynchrony.

To evaluate its effectiveness, a new dataset IRregular Vehicle-to-Vehicle (IRV2V) was created that simulates various real-world scenarios. Extensive experiments on both this new dataset and an existing dataset DAIR-V2X showed that CoBEVFlow consistently outperforms other baseline methods (evaluated on Vehicle detection).

**Strengths:**

1. The challenge of temporal asynchrony in multi-agent collaborative perception is quite new and the proposed CoBEVFlow firstly addresses it by reassigning asynchronous perceptual features (at RoI level) to ensure better alignment.
2. The creation of the IRV2V dataset to simulate real-world scenarios provides an important resource for future research.
3. Detailed evaluations of two datasets provide good results (Fig. 4) for the superiority of CoBEVFlow over existing methods.
4. With smaller bandwidth, Fig.5 shows CoBEVFlow actually shows the potential to greatly improve the performance over single-object detection.



**Weaknesses:**

1. The main concern is the limited evaluation. There might be limitations in applying CoBEVFlow to other types of multi-agent systems as it was mainly tested on vehicle-to-vehicle communication scenarios. Is there any reported performance of Pedestrian or Cyclist?
2. A background paragraph (Sec.3) about the comparison between collaborative and original settings (perhaps a visual comparison rather than a tedious explanation in text) could be more elaborate for better understanding for new readers. Even if I was working on the 3D detection, I cannot understand the key difference quickly.
3. While CoBEVFlow shows potential in simulated scenarios, its performance in broader, real-world situations remains untested.



**Questions:**

1. Will the dataset be released for future research?

**Limitations:**

There is a discussion about limitations.

---

> ### Author Rebuttal · Authors · 2023-08-09
>
> ## W1: Limited Evaluation and Applicability
> 1. Besides vehicle-to-vehicle collaboration communication, we also report the result of experiments on the vehicle-to-infrastructure dataset DAIR-V2X in the main text. This dataset includes information from both the vehicle side and the roadside units, enabling communication between vehicles and infrastructure. Figure.4(b) in the main text and Table 4 in the appendix show that our approach -- CoBEVFlow outperforms the best methods by 5.73% and 2.04% in terms of AP@0.50 and AP@0.70, respectively, under a 300ms interval expectation, which exhibits commendable performance on the vehicle-infrastructure collaborative dataset in real-world scenarios. The visualization of detection results can be referenced in Figure 8 in the Appendix. All these results suggest that our method can effectively handle arbitrary multi-agent systems and exhibits superior efficacy in the real-world setting.
> In addition to the two datasets mentioned in the main text, based on suggestions from other reviewers, we also conducted supplementary experiments on the **V2XSet** dataset. V2XSet is a simulated vehicle-to-infrastructure dataset, which does not consider asynchrony issues. The experimental setup was consistent with the other datasets. The specific results can be seen in Table 1 of the global response PDF.CoBEVFlow outperforms the best SOTA methods by 8.74%, and 17.04% for AP@0.50 and AP@0.70 respectively under the expectation of time intervals is 300ms. The findings obtained from experiments on the V2XSet dataset align with the conclusions drawn from experiments on the other two datasets in this paper.
> 2. Considering more object categories in the evaluation of multi-agent perception systems is a valuable suggestion. The two datasets reported in the main text – the IRV2V dataset and the collaborative portion of the DAIR-V2X dataset, as well as the additional V2XSet dataset provided in the rebuttal, all of them do not include labels for Pedestrians and Cyclists. Consequently, we cannot present corresponding detection results. Moreover, as far as we know, none of the existing collaborative detection datasets currently have annotations for these two entities.
>
> ---
>
> ## W2: Comparing Collaborative and Original Settings
> We appreciate the reviewer's suggestion!
> 1. In the future version, we will provide a more detailed description of collaborative sensing.
> 2. The difference between original and collaborative settings is: Under the original setting, the ego agent can only use the information obtained from its own sensors for detection. However, in a collaborative setting, **different agents can transmit and share information, meaning the ego agent can use information from other agents for detection in addition to its own information**, thereby expanding and enhancing its field of perception. By leveraging complementary perceptual information through communication, agents overcome the inherent limitations of single-agent perception, such as occlusion and long-range issues. In Figure 4 of the main text, the gray dashed lines represent individual perception, while the solid lines represent collaborative perception methods. Under the condition of zero expectation of time interval, the performance of collaborative perception methods is better than that of single perception. However, as the degree of temporal asynchrony increases, the performance of methods like late fusion, V2VNet, V2X-ViT, Disconet, and Where2comm are affected by temporal asynchrony and even fall below the level of single perception. This is precisely the problem we aim to address. As shown by the red line in Figure 4, in the case of temporal asynchrony, our objective is to maintain the performance of collaborative perception at a higher level as much as possible.
>
> ---
>
> ## W3: Real-World Performance of CoBEVFlow
> In the main text, the proposed method is validated on two datasets, our new dataset IRV2V and DAIR-V2X. We believe the experiment results on the DAIR-V2X should reveal the potential of CoBEVFlow in real-world scenarios. DAIR-V2X is the first real-world collaborative perception dataset, which is collected from real road scenes. In this dataset, there is one vehicle and one roadside unit. The communication process takes place between the vehicle and the roadside unit. The dataset is collected from real-world scenarios, encompassing a wide range of conditions including clear, rainy, and foggy days, as well as daytime, nighttime, urban roads, and highways. It is extensively employed in collaborative perception research. Figure.4(b) in the main text and Table 4 in the appendix show that our approach -- CoBEVFlow outperforms the best methods by 5.73% and 2.04% in terms of AP@0.50 and AP@0.70, respectively, under a 300ms interval expectation. In Figure 4(b), the red line represents our method, CoBEVFlow, and it can be observed that our method consistently outperforms others. In Appendix Figure 8, we present the visualization of detection results on the DAIR-V2X dataset with an expected interval of 300ms. The red boxes indicate the predicted results, while the green boxes represent the ground truth. In the presence of temporal asynchrony, our method's predictions closely match the ground truth, resulting in fewer false positives. These results demonstrate that CoBEVFlow continues to achieve superior performance in real-world scenarios.
>
> ---
>
> ## Q1: Dataset Availability for Future Research
> **Yes**, the dataset and the code both will be fully accessible to support research in the field.

---

> > ### Comment · Reviewer_ZUTg · 2023-08-14
> > **Thanks for author's rebuttal**
> >
> > The rebuttal has addressed my concerns. I raised the rating to weak accept.

---

### Author Rebuttal · Authors · 2023-08-09

Thank you for your valuable feedback!

In this work, we propose CoBEVFlow, an asynchrony-robust collaborative 3D perception system based on bird’s eye view flow, to address the issues caused by temporal asynchrony among agents.

In the main text, we conducted experiments on two datasets: IRV2V and DAIR-V2X[1]. IRV2V is the first synthetic collaborative perception dataset with various temporal asynchronies that simulate different real-world scenarios that we created, containing 2 to 5 vehicles as collaborative agents in each frame. **DAIR-V2X is a real-world dataset** consisting of a roadside unit and a vehicle as collaborative agents in each frame. Our experimental results conducted on both datasets are depicted in Figure 4. The corresponding numerical outcomes are detailed in Appendix Table 4 and Table 5. In the case of the IRV2V dataset, CoBEVFlow demonstrates substantial performance, achieving 83.1% and 75.7% in AP@0.50 and AP@0.70 respectively, within the 300ms time interval. This performance exceeds that of the leading baseline method by 23.3% and 35.3%. Similarly, on the DAIR-V2X dataset, CoBEVFlow achieves remarkable results of 73.8% and 59.9% in AP@0.50 and AP@0.70 respectively, within the 300ms time interval, outperforming the best baseline method by 5.73% and 2.04%.

Furthermore, during this rebuttal process, we extended our experiments to include the V2XSet[2] dataset. V2XSet is a simulated dataset where each scenario involves at most one roadside unit and 2 to 4 vehicles as collaborative objects. The outcomes of these experiments are summarized in Table 1 of the global response PDF. Under time delays of 300ms and 500ms, the AP@0.70 scores achieved by CoBEVFlow are 0.776 and 0.713 respectively. These values surpass the best baseline methods by **17.0%** and **10.6%** respectively, and are notably higher than the results of single-object detection(0.556). This once again underscores CoBEVFlow's ability to maintain high levels of collaborative perception performance in scenarios involving temporal asynchrony.

The attached PDF contains various tables and figures. Please review it.

[1] Haibao Yu, Yizhen Luo, Mao Shu, Yiyi Huo, Zebang Yang, Yifeng Shi, Zhenglong Guo, Hanyu Li, Xing Hu, Jirui Yuan, and Zaiqing Nie. DAIR-V2X: A large-scale dataset for vehicle-infrastructure cooperative 3d object detection. In IEEE/CVF Conference on Computer Vision and Pattern Recognition, CVPR 2022, New Orleans, LA, USA, June 18-24, 2022, pages 21329–21338. IEEE, 2022.

[2] Runsheng Xu, Hao Xiang, Zhengzhong Tu, Xin Xia, Ming-Hsuan Yang, and Jiaqi Ma. V2x-vit: Vehicle-to-everything cooperative perception with vision transformer. In Computer VisionECCV 2022: 17th European Conference, Tel Aviv, Israel, October 23–27, 2022, Proceedings, Part XXXIX, pages 107–124. Springer, 2022.

---

### Comment · Area_Chair_QvP8 · 2023-08-13

Dear Reviewers,

Thank you for reviewing this paper. Authors have provided their rebuttal. Would you please check it, and give your comments/rating based on the rebuttal letter and the comments from other reviewers?

Best Regards

AC

---

### Decision · Program_Chairs · 2023-09-21

**Decision:**

Accept (poster)

**Comment:**

In order to address the issues caused by temporal asynchrony among agents, this paper introduces a novel asynchrony-robust collaborative 3D perception framework, based on bird’s eye view flow. All the four reviewers recommend accept, and they pointed out that the task is important and new, and the proposed method is interesting and effective. The authors also did comprehensive evaluations. Though there are some weaknesses such as limited efficiency, considering the novelty of the task and method, the AC recommends accept.